Polycystic ovarian syndrome (PCOS) and recurrent spontaneous abortion (RSA) are associated with the PI3K-AKT pathway activation

Lin Wenjing 1
Wang Yuting 2
Zheng Lei 3 zhenglei@sysush.com
1 Reproductive Medicine Center, The Seventh Affiliated Hospital, Sun Yat-sen University , Shenzhen , China
2 Anesthesiology Department, Shenzhen People’s Hospital, First Affiliated Hospital of Southern University of Science and Technology , Shenzhen , China
3 Anesthesiology Department, The Seventh Affiliated Hospital, Sun Yat-sen University , Shenzhen , China
Guan Fanglin
Electronic publication date: 2024 Sep 6
Publication date: 2024
Volume: 12
Electronic Location ID: e17950
Received 2024 Jun 21; Accepted 2024 Jul 30
Copyright: © 2024 Lin et al.
Copyright year: 2024
Copyright holder: Lin et al.
License: This is an open access article distributed under the terms of the Creative Commons Attribution License, which permits unrestricted use, distribution, reproduction and adaptation in any medium and for any purpose provided that it is properly attributed. For attribution, the original author(s), title, publication source (PeerJ) and either DOI or URL of the article must be cited.
License URL: https://creativecommons.org/licenses/by/4.0/

Keywords: Polycystic Ovary Syndrome (PCOS), Recurrent spontaneous abortion (RSA), “WGCNA”, GSEA, PI3K-AKT signaling pathway, GSEA analysis

Funding: The authors received no funding for this work.

==============================
Aims

We aimed to elucidate the mechanism leading to polycystic ovarian syndrome (PCOS) and recurrent spontaneous abortion (RSA).

Background

PCOS is an endocrine disorder. Patients with RSA also have a high incidence rate of PCOS, implying that PCOS and RSA may share the same pathological mechanism.

Objective

The single-cell RNA-seq datasets of PCOS (GSE168404 and GSE193123) and RSA GSE113790 and GSE178535) were downloaded from the Gene Expression Omnibus (GEO) database.

Methods

Datasets of PSCO and RSA patients were retrieved from the Gene Expression Omnibus (GEO) database. The “WGCNA” package was used to determine the module eigengenes associated with the PCOS and RSA phenotypes and the gene functions were analyzed using the “DAVID” database. The GSEA analysis was performed in “clusterProfiler” package, and key genes in the activated pathways were identified using the Kyoto Encyclopedia of Genes and Genomes (KEGG) analysis. Real-time quantitative PCR (RT-qPCR) was conducted to determine the mRNA level. Cell viability and apoptosis were measured by cell counting kit-8 (CCK-8) and flow cytometry, respectively.

Results

The modules related to PCOS and RSA were sectioned by weighted gene co-expression network analysis (WGCNA) and positive correlation modules of PCOS and RSA were all enriched in angiogenesis and Wnt pathways. The GSEA further revealed that these biological processes of angiogenesis, Wnt and regulation of cell cycle were significantly positively correlated with the PCOS and RSA phenotypes. The intersection of the positive correlation modules of PCOS and RSA contained 80 key genes, which were mainly enriched in kinase-related signal pathways and were significant high-expressed in the disease samples. Subsequently, visualization of these genes including PDGFC, GHR, PRLR and ITGA3 showed that these genes were associated with the PI3K-AKT signal pathway. Moreover, the experimental results showed that PRLR had a higher expression in KGN cells, and that knocking PRLR down suppressed cell viability and promoted apoptosis of KGN cells.

Conclusion

This study revealed the common pathological mechanisms between PCOS and RSA and explored the role of the PI3K-AKT signaling pathway in the two diseases, providing a new direction for the clinical treatment of PCOS and RSA.

Introduction

PCOS is one of the most prevailing endocrine and metabolic disorders that occurs to about 6–20% women at reproductive age (Siddiqui et al., 2022) and is also the primary cause of menstrual complications to females (Chaudhari, Mazumdar & Mehta, 2018). The diagnostic criteria of PCOS include ovulation abnormalities (oligo- or an-ovulation), biochemical hyperandrogenism and the presence of polycystic (12 or more small bubbles) or enlarged ovaries (volume > 10 mL) in ultrasound images (Wang & Mol, 2017), persistent hormonal imbalance and irregular menstruation, which could ultimately lead to infertility (Zehravi, Maqbool & Ara, 2021). Obesity, stress, unhealthy diet and lifestyle and infectious mediators will increase the risk of PCOS (Department of Obstetrics & Gynecology, 2023). Multiple studies reported the etiopathogenesis of PCOS, for example, the defect of ovarian cells (theca cells) caused excessive androgen synthesis (Itriyeva, 2022) and elevated ratio of luteinizing hormone (LH) and follicle-stimulating hormone (FSH) (Raj & Talbert, 1984; Marx & Mehta, 2003). Currently, treatment strategies for PCOS include oral contraceptives, anti-androgen drugs, and metformin. In addition, a number of surgical treatments and clinical strategies such as ovarian perforation and in vitro fertilization have also been used to treat patients with PCOS (Che et al., 2021; Bednarska & Siejka, 2017). However, these medications and surgical procedures are sometimes ineffective and have some side effects. Thus, discovering validated indicators or pathogenic mechanisms has crucial significance for the treatment and prognosis of patients with PCOS.

As a serious complication of pregnancy, recurrent spontaneous abortion (RSA) is defined as having two or more spontaneous abortions with the same partner before 24 weeks of gestation (Practice Committee of the American Society for Reproductive Medicine, 2020). The incidence of RSA in all couples planning to conceive is 1% to 2% (Bender Atik et al., 2023). Woman over 30 years of age have the highest risk of RSA and poorer prognosis, moreover, the risk of RSA increases with the number of miscarriage number, which causes more damage to maternal endometrium and the occurrence of pelvic inflammatory disease and might eventually lead to secondary infertility (Hennessy et al., 2021; Pandey, Rani & Agrawal, 2005). At present, the exact cause of 50% RSA cases is still unclear and these cases are clinically defined as unexplained RSA (Gao & Wang, 2015). Studies showed that 58% patients with RSA might also have PCOS. High levels of luteinizing hormone, hyperandrogenism and hyperinsulinemia would reduce egg quality and endometrial tolerance (Eivazi et al., 2023; Yang et al., 2018), indicating that PCOS and RSA may share a common pathological mechanism.

This study discussed potential pathogenic mechanisms and therapeutic interventions between PCOS and RSA based on the data from public databases. WGCNA was performed to identify gene modules significantly positively and negatively correlated with PCOS and RSA. Subsequently, gene enrichment analyses were performed to analyze the biological processes of gene modules related to PCOS and RSA. Genes significantly expressed in the development of the disease were identified, followed by determining the key enriched signaling pathways and relevant drugs that could inhibit the activation of the pathways. Finally, the expression levels and functions of the key genes were examined using in vitro cellular experiments. In conclusion, the present study systematically investigated the common pathomechanisms of PCOS and RSA, providing potential new targets for the pharmacological interventions in treating the two diseases.

Materials and Methods

Data acquisition and pre-processing

The datasets of PCOS (GSE168404 and GSE193123) were downloaded from the Gene Expression Omnibus (GEO, https://www.ncbi.nlm.nih.gov/geo/) database (Hu et al., 2022). The samples were obtained from the granulosa cells in ovarian tissues. GSE168404 contains five PCOS and five control samples, while the GSE193123 contains three PCOS and three control samples. Similarly, the datasets GSE113790 and GSE178535 for recurrent spontaneous abortion (RSA) samples each contained a total of three RSA and three control samples, and the samples derived from the decidual tissue. The ComBat function in “sva” R package was used to remove the batch effect between samples (Chen et al., 2019).

WGCAN analysis

WGCNA was performed using the “WGCAN” R package (Chen et al., 2019). Based on unsupervised analysis of clusters, a clustering tree example was created to remove any anomalies. Next, Pearson’s analysis was utilized to compute the correlation coefficient among the genetic components for producing a matrix to represent similarities. A scale-free topology criterion of 0.85 between the average annual connectivity (k) and p (k) of the modules was used to determine the optimal soft threshold β to construct a scale-free topology network (the optimal soft threshold for PCOS and RSA queues was 10). Subsequently, the calculation of the topological overlap matrix (TOM) was performed with the adjacency matrix as a foundation, based on which network connectivity among the genes was determined. Hierarchical clustering was performed based on the dissimilarity among the genes to identify co-expression modules. Additional hierarchical clustering of these modules was performed according to the dissimilarity of their module eigengenes, followed by merging the modules with similar patterns of co-expression.

Function enrichment analysis

The Database for Annotation, Visualization and Integrated Discovery (DAVID, https://david.ncifcrf.gov/) database was used for gene function annotation. Significant biological processes were visualized (p < 0.05) (Dennis et al., 2003).

Gene set enrichment analysis

The logFC value of genes in PCOS, RSA and control samples were calculated and ranked from the largest to smallest. Then, the gseGO function in the “clusterProfiler” R package was used to perform the gene set enrichment analysis (GSEA) (Yu et al., 2012) (setting ont = “BP”, OrgDb = org.Hs.eg.db, keyType = “SYMBOL”, minGSSize = 10, maxGSSize = 500, pvalueCutoff = 0.05 and pAdjustMethod = “BH”). The results were visualized applying the gseaNb function in the “GseaVis” R package (Liu et al., 2023).

KEGG pathway analysis

The gene sets of interest were uploaded into the “KEGG Mapper-Color” module in the Kyoto Encyclopedia of Genes and Genomes (KEGG, https://www.kegg.jp/) database (setting red mark) to download the KEGG pathway plot (Kanehisa & Sato, 2020).

Identifying the compounds that inhibited interest genes expression

The genes of interest were uploaded into the Comparative Toxicogenomics Database (CTD, http://ctd.mdibl.org) to identify the compounds that could inhibit the expression of the genes (Davis et al., 2023).

Cell culture and transfection

Human normal ovarian epithelial cell (IOSE80) and human ovarian granulosa cell tumor (KGN) were purchased from Typical Culture Reserve Center of China (Shanghai, China). All the cells were cultured in DMEM (Gibco, Waltham, MA, USA) added with fetal bovine serum (Gibco, Waltham, MA, USA) and penicillin/streptomycin at 37 °C in 5% CO2. The negative control (Vector) and si-RNA (Sagon, China) were transfected into cells using Lipofectamine 2000 (Invitrogen, Waltham, MA, USA). The target sequence of siRNA was shown in Table 1.

Table 1 Target sequence for PRLR.

Gene	Target sequence (5′-3′)	
siPRLR#1	AGCCAACATGAAGGAAAATGTGG	
siPRLR#2	AGCAGTTTCTCGGATGAACTTTA	
siPRLR#3	CAGCAAACAGAGTTTAAGATTCT	

Real-time quantitative PCR

TRIzol (Thermo Fisher Scientific, Waltham, MA, USA) reagent was used to extract total RNA from IOSE80 and KGN cell lines. The cDNA was extracted from 500 ng RNA with the use of HiScript II SuperMix (Vazyme, Nanjing, China). PCR amplification conditions were set at 94 °C for 10 min (min), at 94 °C for 10 s (s), at 60 °C for 45 s. GAPDH was an internal reference. Table 2 displayed a list of primer pair sequences for targeted genes.

Table 2 Primers of genes.

Genes	Forward primer sequence (5′-3′)	Reverse primer sequence (5′-3′)	
PDGFC	TGAACCAGGGTTCTGCATCCAC	TAAGCAGGTCCAGTGGCAAAGC	
GHR	GCAGCTATCCTTAGCAGAGCAC	AAGTCTCTCGCTCAGGTGAACG	
PRLR	CATGGTGACCTGCATCTTTCCG	GTGGGAGGAAAGTCTTGGCATC	
ITGA3	GCCTGACAACAAGTGTGAGAGC	GGTGTTCGTCACGTTGATGCTC	
GAPDH	GTCTCCTCTGACTTCAACAGCG	ACCACCCTGTTGCTGTAGCCAA	

Cell viability

Cell viability was detected using cell counting kit-8 (CCK-8, Beyotime, China). Differently treated cells were cultured in 96-well plates at a density of 1 × 103 cells per well. The CCK-8 solution was applied at specified time point. After incubation at 37 °C for 2 h, the OD 450 value of each well was detected using microplate reader (Thermo Fisher Scientific, Waltham, MA, USA).

Transwell

Transwell assay was performed to assay migration on KGN cell line. Simply, the cells (5 × 104) are inoculated into a cell chamber that is not coated with Matrigel (for migration). Serum free medium was added to the upper layer and complete DMEM medium was added to the lower layer. After 24 h of culture, the migrating cells were fixed with 4% paraformaldehyde for 30 min and stained with 0.1% crystal violet for 20 min. A cotton swab was used to gently wipe off the upper layer of non-migrated cells and wash 3 times with PBS. The cells were then observed under microscope and counted.

Flow cytometry

Apoptosis was analyzed using flow cytometry according to the manufacturer’s operating procedures. Specifically, the cells were harvested with trypsin and then re-suspended in PBS at a concentration of 1 × 104 per 200 μL and stained with Annexin V-FITC and PI solution on ice away from light for 30 min. After rinsing in PBS, the cells were examined using BD FACS Calibur flow cytometry (BD, Franklin Lakes, NJ, USA).

Statistical analysis

The Wilcoxon rank-sum test was used to evaluate the difference between two sets of continuous variables. All statistical analysis and data visualization were performed in the R software (version 4.3.1). Experimental data from at least three repeated experiments were analyzed using GraphPad Prism 9.0 (La Jolla, CA, USA). Statistical comparisons were determined using Student’s t-test and one-way ANOVA. A p < 0.05 was considered as statistically significant.

Results

Identifying gene modules closely associated with the PCOS and RSA phenotypes

After data preprocessing and removal of the batch effect, the WGCNA was used to construct gene co-expression module network and identify the modules related to PCOS and RSA phenotypes. The soft threshold and module clustering were shown in (Fig. S1). In the PCOS dataset, the MEplum1, MEgreen, MEsteelblue and MEdarkgreen modules were significantly positive correlated with the PCOS phenotype, whereas the MEmediumpurple3, MEdarkolivegreen and MEmagenta modules were significantly negative correlated with the PCOS phenotype (Fig. 1A). In the RSA dataset, the MEmidnightblue and MEgreen modules were significantly positively correlated with the RSA phenotype, whereas the MEbisque4 and MEpurple modules were significantly negatively correlated with the RSA phenotype (Fig. 1B).

Figure 1 WGCNA for identifying PCOS and RSA phenotype-related gene module.

(A) The correlation between the co-expression modules and the PCOS phenotype. (B) The correlation between the co-expression modules and the RSA phenotype.

Function enrichment analysis of gene modules related to PCOS and RSA

Subsequently, the function enrichment analysis of gene modules related to PCOS and RSA were performed based on the DAVID database. It was found that the angiogenesis, TGF-β, Wnt and cell regulation pathways were enriched in the gene modules positively correlated with PCOS (Fig. 2A), while the cholesterol, fatty acid, carbohydrate metabolism, phosphorylation and autophagy regulation pathways were enriched in the gene modules negatively correlated with PCOS (Fig. 2B). In addition, the modules positively correlated with RSA were closely associated with the cell division, viral resistance, angiogenesis and Wnt signaling pathways (Fig. 2C), while the cytoplasmic translation, ribosome assembly and cell growth pathways were significantly enriched in the gene modules negatively correlated with the RSA (Fig. 2D).

Figure 2 Function enrichment analysis of gene module.

(A) The function enrichment analysis of PCOS positive correlated gene module. (B) The function enrichment analysis of PCOS negative correlated gene module. (C) The function enrichment analysis of RSA positive correlated gene module. (D) The function enrichment analysis of RSA negative correlated gene module.

Identifying the biological processes closely associated with the PCOS and RSA phenotypes

The results of gene function enrichment analysis demonstrated that the angiogenesis, cell cycle regulation and Wnt signaling pathways were closely associated with the gene modules positively correlated with PCOS and RSA. We first calculated gene module enrichment scores using the ssGSEA method and observed a significant positive correlation between the gene module MEgreen and Wnt signaling pathway (cor = 0.74, p < 0.01). MEplum1 had a significant positive correlation with chromosome segregation (cor = 0.68, p < 0.01) in addition to the Wnt signaling pathway (cor = 0.71, p < 0.01) (Fig. S2A). In the RSA dataset, a significant positive correlation between the MEmidnightblue module and angiogenesis (cor = 0.64, p < 0.05) was detected (Fig. S2B). Furthermore, we preformed the GSEA on these pathways in the PCOS and RSA samples, and the results showed that the POSC and RSA phenotype were significantly positively correlated with the angiogenesis, cell cycle regulation and Wnt signaling pathways (Figs. 3A–3F), suggesting that these pathways played a crucial role in promoting PCOS and RSA phenotypes.

Figure 3 The gene set enrichment analysis (GSEA) for PCOS and RSA related biological process identifying.

(A) The biological process of angiogenesis in PCOS samples. (B) The biological process of the Wnt signaling pathway in PCOS samples. (C) The biological process of chromosome segregation in PCOS samples. (D) The biological process of angiogenesis in RSA samples. (E) The biological process of positive regulation of Wnt signaling pathway in RSA samples. (F) The biological process of positive regulation of MAPK cascade in RSA samples.

Identifying the shared genes between the PCOS and RSA phenotypes

The gene intersection between the RCOS and gene modules positively correlated with RSA included 80 key genes (Fig. 4A), while 13 key genes were obtained from the gene intersection between RCOS and gene modules negatively correlated with RSA (Fig. 4B). However, the enrichment analysis did not annotated the Gene Ontology (GO) terms of these 13 key genes, but the 80 key genes were closely associated with the BPs of the Rho protein signal transduction, Janus kinase activation and the positive regulation of protein kinase B pathways (Fig. 4C). Considering the important role of the kinase activity in cellular signal transduction, the expression of these kinase-related genes in the disease and control samples were evaluated. It was found that most genes were significantly expressed in the disease samples except the PLEKHGS genes such as ITGA3, PLEKHG5 and RHOB (Figs. 4D, 4E).

Figure 4 Identifying the shared genes between PCOS and RSA samples.

(A) Venn plot of the intersection between the PCOS positive correlated genes and RSA positive correlated genes. (B) Venn plot of the intersection between the PCOS negative correlated genes and RSA negative correlated genes. (C) The biological process of 80 positive correlated genes. (D) The expression of kinase-related genes in PCOS samples. (E) The expression of kinase-related genes in RSA samples. *p < 0.05, **p < 0.01, ***p < 0.001.

The activation of PI3K-AKT pathway was closely associated with the PCOS and RSA phenotypes

The pathway analysis of KEGG revealed that these kinase genes, such as the PDGFC encoding growth factor (GF) protein, the GHR and PRLR encoding CytokineR protein, played crucial role in PI3K-AKT signaling pathways (Fig. 5). After activation, these proteins all signaled to the PI3K, leading to cascade reactions. Therefore, the PI3K-AKT pathway may be a key pathway in promoting PCOS and RSA occurrence.

Figure 5 Kinase-related genes in the PI3K-Akt pathway.

Identifying the compounds associated with kinase inhibition

We downloaded and filtered the drugs that could inhibit the expression of PDGFC, GHR, PRLR and ITGA3. The results showed that bisphenol A, Benzo(a)pyrene, Tetrachlorodibenzodioxin and tobacco smoke pollution can be effective drugs to inhibit these genes in the PI3K-AKT pathway (Fig. 6), and that these compounds were considered as the candidate drugs for PCOS and RSA treatment and intervention.

Figure 6 Venn plot of kinase-inhibited drugs.

Experimental validation

The expression levels of PDGFC, GHR, PRLR and ITGA3 in IOSE80 and KGN cell lines were measured using RT-qPCR. The results showed that only the transcriptional level of PRLR was elevated in KGN cells (p < 0.001, Figs. 7A–7D). Next, analysis on the inhibition efficiency of siRNA in KGN cells demonstrated that siPRLR#3 had the highest efficiency, therefore it was used in the following experiments (p < 0.01, Fig. 7E). Functionally, knockdown of PRLR suppressed cell viability compared to siNC group (Fig. 7F). Cell migration was inhibited by knockdown PRLR in KGN cells (Figs. 7G, 7H). The apoptosis rate was significantly promoted by loss of PRLR (Figs. 7I, 7J). Those data revealed that PRLR may contribute to PCOS.

Figure 7 Validation of experimental.

(A–D) Transcription levels of PDGFC, GHR, PRLR, and ITGA3 in IOSE80 and KGN cell lines were detected by RT-qPCR, and relative quantitative analysis was performed. (E) RT-qPCR was used to verify the inhibition efficiency of small interfering RNA. (F) Changes of cell viability of KGN cells in normal group and KGN cells inhibited by PRLR. (G, H) Changes in cell migration ability of KGN cells in normal group and KGN cells with inhibition of PRLR. (I, J) Changes of apoptotic ability of KGN cells in normal group and KGN cells that inhibit PRLR. ns: no significance, **p < 0.01, ***p < 0.001, ****p < 0.0001.

Discussion

RSA is a pregnancy-associated complication of the endocrine malfunction disease PCOS (Alkhuriji et al., 2020). Insulin resistance, obesity, hyperhomocysteinaemia, hyperinsulinaemia, hyperandrogenaemia and poor endometrial receptivity are causal factors to the development of PCOS (Pluchino et al., 2014), which are also reported in RSA woman. However, whether the two diseases shared a common mechanism that promoted the disease development remained unclear. Hence, the current work explored the mechanism underlying the occurrence of PCOS and RSA. This study collected the RNA-seq data of PCOS and RSA samples and performed WGCNA to identify the gene modules closely associated with PCOS and RSA phenotypes. The results revealed that the module genes in PCOS and RSA were all enriched in angiogenesis, Wnt and cell cycle regulation pathways. The intersection of these genes showed that the PI3K-AKT signaling pathway played a critical role in promoting the occurrence of PCOS and RSA. Importantly, these genes were validated using in vitro cellular models and may have potential clinical implications for the treatment of patients with PCOS and RAS.

In some extreme cases of PCOS, ovarian stroma hyperthecosis and polycystic ovaries with disturbed angiogenesis are observed (Meczekalski et al., 2021), characterized by increased vascularization of ovarian stroma (Delgado-Rosas et al., 2009). The transplantation of placenta-derived mesenchymal stem cells (MSCs) promotes follicular development, restores the function of ovarian through vascular remodeling by releasing VEGF (Cho et al., 2021), and regulates the immune balance of maternal-fetal interface for RSA treatment and intervention (Deng, Liao & Zhu, 2022; Shafei et al., 2022). VEGF antagonists have been developed to inhibit angiogenesis and reduce the vascular network of cysts to prevent the growth of cysts (Hull et al., 2003; Chu et al., 2017). The study of animal model showed that the local platelet-derived growth factor B injection can lower the VEGF concentration in intraovarian and partially re-establish the number of small follicles, thereby limiting the formation of cystic follicle (Di Pietro et al., 2016). Local quantitative alterations of fibrillin-3, perlecan, collagen type IV and proteins like pentraxin-3 (PTX3) are correlated with the PCOS phenotype (Ma et al., 2020). More specifically, increased collagen deposition of ovarian stroma, which could aggravate tissue fibrosis, is considered as a hallmark of PCOS (Papachroni et al., 2010). Though the exact mechanisms of increased collagen deposition remained unclear, previous evidence showed that TGF-β RI inhibitor could significantly reduce the pro-fibrotic TGF-β and increase the anti-fibrotic MMP2 level to prevent collagen deposition in a rat model of PCOS (Wang et al., 2018), indicating that the TGF-β signaling pathway may play a key role in the disease etiology (Wang et al., 2018). Above findings suggested that the dysregulation of these pathways might be able to promote PCOS and RSA occurrence, and that the mechanisms of PCOS and RSA involve multiple common pathological pathways, including angiogenesis, TGF-β, Wnt, and cell cycle regulatory pathways. In a word, these results emphasized the potential of targeting these pathways as an effective intervention in the treatment of PCOS and RSA, providing a scientific basis for future clinical treatment strategies.

Currently, many studies revealed the important role of the PI3K-Akt pathway in glycogen and protein synthesis and the regulation of cell division, growth and proliferation, senescence and death (Jafari et al., 2019; Very et al., 2018; Xu et al., 2022). In addition, other study demonstrated that the PI3K-Akt signaling pathway also played a crucial role in promoting the development of ovarian. Premature ovarian insufficiency (POI) is an ovary degenerative disease that affects 1% fertile woman under 40 years old and causes infertility (Cohen, Chabbert-Buffet & Darai, 2015; Gleicher, Kushnir & Barad, 2015). POI occurs when the primordial follicles pool is depleted or inhibited during activation (Kawashima & Kawamura, 2017). Clinically, POI is correlated with increased gonadotropin production and low ovarian steroids (Kozub et al., 2017; Ishizuka et al., 2021). One of the putative mechanisms of POI is the disturbances of Hippo and Akt signaling-transduction pathways (de Koning et al., 2008). Past study using a POI mouse model revealed that the mouse with a lack of a major negative regulator of PI3K has Akt hyperactivation in initial follicles (Reddy et al., 2008). This results in a massive and uncontrollable recruitment of the follicles, followed by increased atresia and an overall diminishment of follicular pool (Reddy et al., 2008). Importantly, the identified high-expressed kinase-related genes (such as PRLR, ITGA3, PLEKHG5 and RHOB) in the PI3K-Akt pathway could be potential targeted genes for disease treatment and intervention. Moreover, our experimental data showed that PRLR influenced KGN cell viability and apoptosis. Similarly, some researchers also reported the role of PRLR in the etiology and proliferation of prolactin (PRL)-induced breast cancer (Kavarthapu, Anbazhagan & Dufau, 2021). Extensive epidemiological research showed that a higher level of circulating PRL is linked with a greater risk of breast cancer and metastasis to premenopausal women (Tworoger & Hankinson, 2008). Notably, Dagil et al. (2012). observed that the binding of PRL and PRLR leads to receptor dimerization, which activates the classical JAK-STAT pathway and some other signaling cascades, including PI3K/AKT and RAF/MEK/ERK. This further supported the importance of the PI3K-Akt signaling pathway in female-associated reproductive diseases, suggesting that the aberrant activation of this pathway is likely to be a common driver of both PCOS and RSA.

However, some limitations in the study should be equally noted. First, the data analyzed only came from the GEO database, and the small sample size of these datasets may not fully represent the diversity and heterogeneity of PCOS and RSA. Future study will integrate more data from different research centers and multiple databases to promote the representativeness and credibility of the results. In addition, this study revealed the potential role of the PI3K/Akt pathway in PCOS and RSA applying bioinformatics analysis but lacked further experimental validation using animal models and clinical samples. Finally, the association between gene expression and specific clinical phenotypes should be analyzed using large-scale clinical data to enhance the clinical application of the current results.

Conclusion

This study performed WGCNA to explore the potential common mechanism associated with the PCOS and RSA phenotypes. The results showed that the PCOS and RSA were all closely associated with the angiogenesis, Wnt and cell cycle regulation pathways. Further analysis demonstrated that a series of kinase-related genes in the PI3K-Akt pathway were significantly high-expressed in the two disease samples, and their targeted drugs were also identified for PCOS and RSA intervention.

Supplemental Information

Supplemental Information 1 Identifying gene modules closely associated with the PCOS and RSA phenotype.

(A) Soft threshold filtering in PCOS data set and the relationship between soft threshold and connectivity. (B) Hierarchical clustering trees in WGCNA analysis within PCOS dataset. (C) Soft threshold filtering in RSA data set and the relationship between soft threshold and connectivity. (D) Hierarchical clustering tree for WGCNA analysis in RSA dataset.

Supplemental Information 2 Correlation analysis based on ssGSEA enrichment scores to explore PSCO and RSA-associated gene modules with critical pathway genes.

(A) Correlation analysis of gene modules significantly positively associated with PCOS (MEplum1, MEgreen, MEsteelblue, and MEdarkgreen) with angiogenesis, WNT signaling pathway, and chromosome segregation. (B) Correlation analysis of gene modules significantly positively associated with RSA (MEmidnightblue and MEgreen) with angiogenesis, WNT signaling pathway, and positive regulation of MAPK cascade.

Supplemental Information 3 MIQE checklist.

Abbreviations

PCOS Polycystic ovarian syndrome

RSA recurrent spontaneous abortion

WGCNA weighted gene co-expression network analysis

GEO Gene Expression Omnibus

DAVID Database for Annotation, Visualization and Integrated Discovery

GSEA Gene Set Enrichment Analysis

KEGG Kyoto Encyclopedia of Genes and Genomes

CTD Comparative Toxicogenomics Database

TGF-β transforming growth factor-beta

GO Gene Ontology

PI3K-Akt Phosphatidylinositol-4,5-bisphosphate3-kinase and serine/threonine kinase Akt

VEGF vascular endothelial growth factor

MSCs mesenchymal stem cells

POI Premature ovarian insufficiency

Additional Information and Declarations

Competing Interests

Author Contributions

Data Availability

The authors declare that they have no competing interests.

Wenjing Lin conceived and designed the experiments, analyzed the data, prepared figures and/or tables, authored or reviewed drafts of the article, and approved the final draft.

Yuting Wang conceived and designed the experiments, performed the experiments, analyzed the data, authored or reviewed drafts of the article, and approved the final draft.

Lei Zheng performed the experiments, analyzed the data, prepared figures and/or tables, and approved the final draft.

The following information was supplied regarding data availability:

The datasets generated and/or analyzed during the current study are available at GEO: GSE19123, GSE168404, GSE113790, GSE178535.

The raw data is available at GitHub and Zenodo:

- https://github.com/6Linwenjing/Updated-new-data.git

- 6Linwenjing. (2024). 6Linwenjing/Updated-new-data: My updated data (v.1.1.0). Zenodo. https://doi.org/10.5281/zenodo.12198919

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
