# Peer review of "Polycystic ovarian syndrome (PCOS) and recurrent spontaneous abortion (RSA) are associated with the PI3K-AKT pathway activation"

_PeerJ, doi:10.7717/peerj.17950_

## Round 0.1 · original submission · Major Revisions

Based on the reviewers' feedback, we have decided that major revisions are necessary. The reviewers have identified several critical areas requiring significant improvement, including the need for a more comprehensive and detailed introduction that clearly explains the significance of linking PCOS and RSA, and the implications for clinical treatment. Additionally, the Methods section, particularly the description of WGCNA analysis, needs to be more detailed, and the cellular experiments should be reflected in the abstract and introduction. The Discussion section should be expanded to provide a thorough exposition of the research findings and their rationalization. Finally, grammatical issues and clarity throughout the manuscript must be addressed. We believe that these revisions will significantly enhance the quality of your manuscript and look forward to your resubmission.

Reviewer 1 ·

Basic reporting

Polycystic ovary syndrome (PCOS) is a common endocrine disease, and patients with recurrent spontaneous abortion (RSA) also show a high prevalence of PCOS, which was used as the starting point of this study to investigate whether PCOS shares the same pathologic mechanism with RSA. In this study, we first downloaded the single-cell RNA-seq dataset of PCOS and the dataset of RSA from the GEO database, and mined the target gene modules by WGCNA. Enrichment analysis revealed that the biological processes of angiogenesis, Wnt and cell cycle regulation were significantly and positively associated with PCOS and RSA phenotypes. Next, key genes affecting the disease were obtained, visualized and analyzed, and their effectiveness in disease progression was verified by cellular experiments. In conclusion, this is a bioinformatics co-experimental study with a poor overall idea, but the following issues still need to be addressed before publication:
1. The introduction section needs to be further deepened, for example, by adding more information about clinical treatment strategies for PCOS and systematically elucidating the strengths and weaknesses of these treatment strategies, and by suggesting what the significance of this study is.
2. The introductory section of this study is not well developed and does not elaborate on what is the significance of the conduct of this study, in particular why PCOS must be linked to RSA at this point in time, and does linking the two have any implications for the treatment of clinical disease? Please provide an explanatory statement.
3. Perhaps PCOS and RSA do have several interrelated triggers, as reflected in the introductory section, but what kind of analyses and experiments were used to unearth these triggers? Do they have implications for the conduct of this study? Please elaborate in the introduction section.
4. This study is an analysis of a joint bioinformatics experiment, but it is not reflected in the abstract section as well as the introduction section that this paper was analyzed in a cellular experiment, thus it is recommended that this be added to the original text, thus making the full text clearer.

Experimental design

5. Since this study is to investigate the mechanisms associated with PCOS and RSA, why, after analyzing the downstream regulatory pathways of the relevant gene modules, were the expression levels of the genes associated with these pathways not analyzed? Should the corresponding analysis be added so as to make this paper more coherent? Please explain and add analysis where necessary.
6. The description related to the WGCNA analysis in the Methods section is too simple, in addition to the soft threshold setting, the WGCNA analysis includes a number of key parameters, and the construction of the adjacency matrix is inclusive of many details, which are not reflected in the Methods section, and thus it is suggested to add the complete text in the original article.

Validity of the findings

7. The Discussion section does not provide a general exposition of the research in this paper, and this is also suggested to be added in the opening section of the Discussion. In addition, several of the findings of this paper were not rationalized in the Discussion section, such as rationalizing the mechanisms linking PCOS and RSA, and thus it is recommended that this be added.
8. The Discussion section illustrates animal model-related studies that elucidate the important role of endothelial growth factor antagonists in inhibiting follicle formation in mice; can this be linked to the conclusions of the present study to reveal the mechanisms involved in PCOS? Please explain the role of endothelial growth factor in PCOS in relation to the findings of this study.

Additional comments

9. This paper proposes that many hormones are released during the pathogenesis of PCOS and that the release of these hormones is not clearly a cause or a consequence of PCOS, but is there any substance to the distinction between the two? How does this relate to therapeutic strategies and mechanistic studies of PCOS? Please provide an explanatory note.
10. The etiology of RSA remains unclear, but is PCOS a cause or a consequence of RSA? Has this been reported in the available literature? It is recommended that this be added to.

Reviewer 2 ·

Basic reporting

In this study, the author explores the potential link between the polycystic ovarian syndrome (PCOS) and recurrent spontaneous abortion (RSA). Prognostic factors and response signatures of these two diseases were identified through bioinformatics methods. The experimental design is rigorous. However, there are still some deficiencies in details in the manuscript. Please revise it carefully according to the comments below.
#1. In the abstract section, please check the Methods and Materials section if this paper uses PCOS single cell dataset.
#2. In the methods of abstract, the “DAVID” database was used to annotate the gene function, but its results are not shown.
#3. In the conclusion of abstract section, what is the prospect or significance of this article.
#4. What are the current treatments for PCOS and RSA.
#5. The PCOS and RSA positively correlated modules were associated with angiogenesis, TGFb pathway, WNT pathway, and cell cycle pathways, what important role does PI3K-AKT play in these pathways.
#6. When performing WGCNA analysis, are the gene sets that characterize the PCOS and RSA phenotypes used, and are they present in the Materials and Methods section.
#7. These factors, such as insulin resistance, obesity, hyperandrogenaemia, hyperinsulinaemia, hyperhomocysteinaemia and poor endometrial receptivity, whether any studies have reported the association between PI3K-AKT and these factors.
#8. Line 182, Is there an extra word of modules in this sentence.
#9. Line 215, there is something wrong with this grammar, please correct it.
#10. What are the prospects and shortcomings of this article.

Experimental design

no comment

Validity of the findings

no comment

---

## Round 0.2 · accepted · Accept

We have carefully considered your revised manuscript along with the reviewers' recommendations. Both reviewers have expressed their satisfaction with the changes made and have recommended acceptance. After thorough evaluation, I am pleased to inform you that your manuscript has been accepted for public

cation.

Reviewer 1 ·

Basic reporting

Polycystic ovary syndrome (PCOS) is a common and complex endocrine and metabolic disorder in women of reproductive age, characterized by chronic anovulation (ovulation dysfunction or loss) and hyperandrogenism (excess production of male hormones in women's bodies). Its main clinical manifestations include irregular menstrual cycles, infertility, hirsutism, and/or acne, making it the most common female endocrine disorder. However, the incidence of polycystic ovary syndrome in patients with recurrent spontaneous abortion (RSA) is 58%. High concentrations of luteinizing hormone, hyperandrogenism, and hyperinsulinemia reduce egg quality and endometrial receptivity. This suggests that PCOS and RSA share a common pathological mechanism. Zheng et al. identified gene modules significantly associated with PCOS and RSA using various bioinformatics methods, and found that both PCOS and RSA were significantly positively correlated with angiogenesis, WNT pathway, and cell cycle. Furthermore, they confirmed that kinase related genes ITGA3, PLEKHG5, RHOB, etc. are significantly overexpressed, and these genes are mainly involved in the regulation of the PI3K-AKT pathway, suggesting that PCOS and RSA are both related to the activation of the PI3K-AKT pathway. Finally, they validated the manuscript through experiments. Overall, the manuscript has a certain degree of innovation, the research is complete, the statistics are appropriate, the experimental design is rigorous, and it meets the publishing standards.

Experimental design

no comment

Validity of the findings

no comment

Additional comments

no comment

Reviewer 2 ·

Basic reporting

no comment

Experimental design

no comment

Validity of the findings

no comment

Additional comments

The author responded well to my question, and I have no new comments left